# Autophagy-Mediated Regulation of Different Meristems in Plants

**DOI:** 10.3390/ijms23116236

**Published:** 2022-06-02

**Authors:** Shan Cheng, Qi Wang, Hakim Manghwar, Fen Liu

**Affiliations:** Lushan Botanical Garden, Chinese Academy of Sciences, Jiujiang 332000, China; chengshan@lsbg.cn (S.C.); wangqi@lsbg.cn (Q.W.)

**Keywords:** autophagy, root meristem, stem meristem, plant stress, plant development

## Abstract

Autophagy is a highly conserved cell degradation process that widely exists in eukaryotic cells. In plants, autophagy helps maintain cellular homeostasis by degrading and recovering intracellular substances through strict regulatory pathways, thus helping plants respond to a variety of developmental and environmental signals. Autophagy is involved in plant growth and development, including leaf starch degradation, senescence, anthers development, regulation of lipid metabolism, and maintenance of peroxisome mass. More and more studies have shown that autophagy plays a role in stress response and contributes to maintain plant survival. The meristem is the basis for the formation and development of new tissues and organs during the post-embryonic development of plants. The differentiation process of meristems is an extremely complex process, involving a large number of morphological and structural changes, environmental factors, endogenous hormones, and molecular regulatory mechanisms. Recent studies have demonstrated that autophagy relates to meristem development, affecting plant growth and development under stress conditions, especially in shoot and root apical meristem. Here, we provide an overview of the current knowledge about how autophagy regulates different meristems under different stress conditions and possibly provide new insights for future research.

## 1. Introduction

Autophagy is an extremely conserved degradation process in which cellular contents are delivered to plant and yeast vacuoles or mammalian lysosomes in order to maintain cellular homeostasis [1,2]. Increasing evidence has revealed that autophagy plays an essential role in plants under abiotic and biotic stress conditions, for instance, drought [3], nutrient starvation [4,5], heating [6], chilling [7], oxidative stress [8], pathogen infection [9,10], hypoxia [11], high salt [3,12], etc. The purpose of maintaining a basal level of autophagy when plants grow under normal conditions is to remove damaged organelles and other substances produced during normal growth and metabolism, maintain normal cell metabolism, and ensure a stable intracellular environment. In many plant species, oil is a primary seed storage reserve. Plant peroxisomes play a key role by hydrolyzing the fatty acids and provides seedlings with essential energy before photosynthesis begins. Autophagy plays a vital role in degrading the peroxisomes during this process [13]. Germination of plants overexpressing *ATG8* (*AUTOPHAGY-RELATED 8*) performs better than the wild-type in *Arabidopsis thaliana* [12]. Based on emerging evidence, autophagy has been observed to be involved in various key steps of plant reproduction [14], such as controlling anther development [15], pollen germination [16], development of sperm cells [17] and embryogenesis [18,19]. Several studies in Arabidopsis, such as those on *atg* mutants, established the relationship between autophagy and the phytohormones’ homeostasis. Transcriptomic and metabolomic analyses revealed SA (Salicylic acid) biosynthesis and SA accumulation in *atg* mutants under low nitrate conditions [20]. Through genetic, pharmacological and biochemical approaches, it was clearly demonstrated that the major factor of *atg*-dependent chlorotic cell death is due to the excessive SA signaling [21].

Plant growth and development process is divided into vegetative growth and reproductive growth, and vegetative growth is the process of embryogenesis: two populations of meristematic cells arise and grow in polar directions throughout the life of the plant. The SAM (shoot apical meristem) produces the aerial part of the plant, while the underground part is produced by the RAM (root apical meristem) [22]. Plants have the ability to regulate their own body plans after embryogenesis which helps them in dealing with the environmental changes that affect normal plant growth. This flexibility in organogenesis and development is facilitated by pools of dividing, pluripotent cells that are located in structures called meristems. The maintenance of meristem activity depends on the dynamic balance between stem cell division and differentiation activities and plays an important role in plant growth and development. While the effect of autophagy on meristems has received less attention, the role of autophagy in mediating plant meristems have been demonstrated by increasing evidence. This review presents our new insights regarding the role(s) of autophagy in various meristems of plants and the implications for further research based on recent advances in the field.

## 2. Autophagy

There are three main types of autophagy on the basis of their mechanism; macroautophagy [23], microautophagy [24], and mega-autophagy (Figure 1). In plants, the following two types of autophagy are known: macroautophagy and microautophagy [25].

### 2.1. Macroautophagy

Macroautophagy is initiated with the formation of cup-shaped membranes, which are involved in packaging the target substrates in the cytoplasm. Then, the contents are degraded by the vacuole for cyclic utilization. Transcription factors (TFs) are important participants in plant growth and response to external stimuli. An increasing number of TFs that regulate autophagy genes have been identified. Arabidopsis *WRKY33* (WRKY DNA-binding protein 33) is the first reported TF that interacts with *ATG18a* and mediates autophagy in plants’ necrotrophic pathogen immunity [26]. Arabidopsis *HY5* (Elongated hypocotyl 5), which can directly bind to the promoters of *ATG5* and *ATG8e*, inhibit their gene expression and negatively regulate autophagy [27]. In addition, signaling response elements downstream of plant hormones have also been found to transcriptionally regulate autophagy gene expression [28]. Tomato *ERF5* (ethylene response factor 5) is a typical drought-responsive TF that positively regulates the expression of *ATG8d* and *ATG18h* by binding to DRE-binding sequence on *ATG8d* and *ATG18h* promoters, thereby participating in ethylene-mediated autophagy [29]. BRs (Brassinosteroids) and their signaling element BZR1 (brassinazole-resistant 1) can also upregulate *ATGs* expression and induce autophagy under cold stress in tomato [7,28]. These results suggest that TFs in hormonal signaling can regulate the expression of *ATGs* to promote autophagy. Moreover, some studies have found that 225 TFs from 35 TF families, which are also involved in plant development and stress response, can bind to the promoters of four *ATG8* genes by yeast one-hybrid (Y1H) screens [30]. However, whether these potential TFs are involved in the plant autophagy signaling pathway remains to be confirmed.

### 2.2. Microautophagy

In animals and yeasts, microautophagy is a process in which the vacuole directly packages target substrates by membrane invagination to create autophagic bodies [31]. By contrast, there is little evidence about how microautophagy proceeds in plants [32]. In plants, microautophagy participates in the accumulation of anthocyanin in the vacuole, degrading cellular components and eliminating damaged chloroplasts during starvation [33]. In contrast, there is the less available information on the molecular mechanism of microautophagy than on the general macroautophagy, because there has been no extensive research on microautophagy. The crucial function of ATG proteins in plant macroautophagy has been quite well described, but the key function in microautophagy remains unknown [34]. A more specific analysis of plant microautophagy may reveal surprising aspects of its contribution to plant development and stress response. In addition, future research on plant microautophagy may lead to improved crop plant growth methods.

### 2.3. Mega-Autophagy

Mega-autophagy is the most extreme form of autophagy. Here, the vacuolar hydrolase will be released directly into the cytoplasm by penetration or rupturing of the tonoplast, where cytoplasmic micromaterials are degraded in situ by them [35]. Mega-autophagy usually represents the final stage of PCD (programmed cell death), which occurs in response to pathogen invasion [e.g., the HR (hypersensitive response)] or during development (e.g., during xylogenesis). Animals also use a vesicle-independent chaperone-mediated autophagic pathway. It uses specialized transporters to introduce substrates directly into vacuoles [36]. There is no evidence yet for this pathway in plants. The chaperone-assisted transporters have been found in animals, but not yet in plants (e.g., LAMP-2A).

## 3. The Role of Autophagy in Plants

### 3.1. Autophagy and Plant Growth and Development

Plants need the participation of autophagy in the normal growth and development process, and autophagy plays an important role in the process of plant seed germination, growth and development, reproduction, and aging. Recent studies reveal the importance of autophagy in various physiological processes in multiple plant species [37,38]. After seed germination, abundant lipidated ATG8 protein was detected, suggesting that autophagy plays a crucial role in regulating endosperm nutrients and promoting early seedling development [39]. Autophagy can degrade substances that inhibit seed germination. ATI1 and ATI2 are plant-specific receptors for selective autophagy cargo that interact with Atg8. In Arabidopsis, lack of *ATI1* and *ATI2* or *ATI1* overexpression delayed or stimulated seed germination after ABA treatment, respectively [40]. In Arabidopsis *atg5* mutants, lower nitrogen dilution, reduced rosette size, and defects in seed filling with carbon were observed, indicating that there are defects in photosynthetic carbon assimilation [41]. In Arabidopsis, autophagy regulates root cells and root hair formation [42]. In the process of plant growth and development, the transformation of some special structures requires the participation of autophagy, such as the formation of xylem, the degradation of stolon and tapetum. *Populus* exhibits both compositional and organizational characteristics of angiosperm secondary xylem, with longitudinally oriented water-carrying vascular elements embedded in numerous radially oriented parenchymatic rays and xylem fibers [43]. Under the combination of cytological, microscopic, and high-throughput gene monitoring techniques, the process of cell death in *Populus* xylem fibers was observed. During lignification, a series of autophagy genes were up-regulated in *Populus*, suggesting that autophagy is involved in xylem maturation [44]. Autophagy is also the key to the germination of tobacco pollen. At the early stage of pollen germination, autophagy activity is significantly increased to degrade the cytoplasm in the germination pore. Autophagy was blocked in tobacco *ATG2*, *ATG5*, and *ATG7* silenced plants, and cytoplasmic degradation of germination pores was also inhibited during pollen germination [16]. *ATG6*, *ATG7*, and *ATG10* control pollen fertility, and when autophagy genes are absent, autophagosomes cannot be formed normally, resulting in reduced pollen quantity and vitality of plants [15]. The *atg* mutant plants have fewer inflorescence branches due to premature senescence, which often results in lower seed yield [25]. Seed maturation and germination require significant nutrient reactivation, and autophagy is involved in these important processes. At the seed maturation stage after pollination, *ATG8*, a key protein of autophagy pathway, is increased in endosperm of maize (*Zea mays* L.) starch, indicating enhanced autophagy activity. Most *atg* mutants had normal embryonic development and vegetative growth, but exhibited a typical autophagy-deficient phenotype, namely premature leaf senescence and sensitivity to nutrient starvation [45,46]. The premature leaf senescence of these mutants can be alleviated by blocking SA biosynthesis or signaling. For example, overexpression of SA hydroxylase to synthesize *NahG* can completely inhibit the progeria phenotype of *atg2* and *atg5*, while application of the SA analog BTH (benzothiadiazole) restored the progeria phenotype of these mutants [41].

### 3.2. Autophagy in Abiotic and Biotic Stresses

Previous studies have suggested that both carbon and nitrogen starvation can induce an increase in autophagy in Arabidopsis. With the deepening of research, it has been revealed that many types of abiotic stresses can induce the production of autophagy in plants, recycle damaged macromolecules and cellular components, maintain cellular homeostasis, and help plants resist external stress. In Arabidopsis, soybean, millet, or rice, the overexpression of *ATG8* family homologous genes promotes plant growth, improves tolerance to nitrogen starvation and drought, and increases yield [12,47,48,49]. Recently, studies have also shown that chlorophagy induced by oversupply of nitrate can inhibit phosphorus starvation in Arabidopsis, indicating that autophagy, as the center of nutrient regulation, comprehensively coordinates the availability of nitrogen and carbon under phosphorus starvation, thereby improving phosphorus deficiency resistance [50]. After submergence stress, induced autophagy can maintain intracellular homeostasis by regulating SA, indicating that autophagy plays a role in the hypoxia response in Arabidopsis [11]. Under salt stress, the autophagy activity of the transgenic apples overexpressing *MATG10* and *MATG8i* was significantly higher than that of the wild type, and the salt tolerance was markedly improved [51]. Cold and heat stress significantly induced the expression of *ATGs* and the formation of autophagosomes [7,52]. The overexpression of *MdATG18a* increased the basal heat tolerance of apple *(Malus domestica*) plants, significantly reduced chloroplast damage, enhanced photosynthesis, and increased antioxidant enzyme activity under high temperature [52]. Heterologous overexpression of *MdATG5-1/2* and *Md ATG18a* in tomato resulted in enhanced autophagy activity and increased drought tolerance compared with wild type after drought stress [49]. Oxidative stress is a type of stress caused by high concentration of ROS (reactive oxygen species) in plant cells, which damages cellular components, such as proteins, lipids, and nucleic acids, and produces toxic effects on cells, thus inhibiting the normal growth of plants. Abiotic and biotic stresses can lead to the overproduction of ROS and RNS (reactive nitrogen species) [53,54,55]. There is an interaction between ROS and autophagy: ROS can induce autophagy, and autophagy can reduce the ROS production. The H_2_O_2_ and O^2−^ were significantly accumulated in apple leaves under drought stress, but less in apple *ATG18a*-overexpressing plants, resulting in enhanced drought resistance in autophagy-overexpressing plants [47]. ROS are also involved in regulating the activity of ATG4 protease during autophagosome formation, and ATG4 regulates the binding of ATG8 to phosphatidylethanolamine (PE) through cleavage and delipidation [56].

Autophagy can also be induced in plants under biotic stress. Autophagy plays a dual role in plant-pathogen interactions: promoting host cell death or survival, depending on the pathogenicity mechanism of the different pathogens [57]. Plant auto-immune systems have also evolved complex defense mechanisms, in which HR, in the form of PCD, plays an important role in limiting pathogen invasion [58]. Autophagy contributes to necrotrophic pathogen immunity in plants. WRKY33 was reported to directly bind to ATG18a to positively regulate necrotrophic pathogen immunity in Arabidopsis. *wrky33* mutants are highly sensitive to the necrotrophic pathogen *Botrytis cinerea*, and *B. cinerea* induced *ATG18a* and autophagy are disrupted in *wrky33* [26]. For necrotrophic pathogens, autophagy is often involved in limiting plant HR responses and increasing resistance. In contrast, autophagy-mediated HR restriction has been shown to increase the infectivity of live pathogens in biotrophic disease resistance [59]. During the infection of *Pseudomonas syringae pv*. tomato DC3000, the necrosis of the leaves of the Arabidopsis *atg* mutant did not spread, and the plants showed obvious resistance [60].

## 4. Meristem

Cell mass with the ability to divide continuously or periodically, located in the growing parts of plants, are called meristems. Root and shoot meristems are produced during embryogenesis [61]. Post-embryonically, branching is formed from AMs (axillary meristems) form in the leaf axil. During reproductive growth, IM (inflorescence meristem) are transformed from SAM for successful plant reproduction in Arabidopsis. FMs (floral meristems) are formed from the IM and generate floral organs, including carpels, sepals, stamens, and petals in Arabidopsis. Unlike IM, FMs only show transient stem cell activity. Secondary growth in gymnosperms and woody dicotyledons is dominated by lateral meristems, which is associated with rhizome thickening and re-formation of protective layers [62].

### 4.1. Root Apical Meristem (RAM)

The root apical meristem is adjacent to the root cap and consists of the root quiescent center (QC) and the root stem cell area, responsible for the formation of the main root. The QC is responsible for maintaining the surrounding stem cells. It also helps to maintain the properties of stem cells. Vascular stem cells are responsible for the formation of vascular tissue, root cap stem cells play a crucial role in forming root cap cells, and cortical/endodermis stem cells help to form cortex and endodermis [63].

TFs and hormones together play important regulatory functions in the root apical meristem. As a key regulator of the quiescent center of the root apical meristem, WOX5 (WUSCHEL RELATED HOMEOBOX 5) has been considered to be important for maintaining the balance of quiescent center cells and column cells [64,65]. The root cap stem cells of the *wox*5 mutant lose their stem cell activity and differentiate into root cap cells. Meanwhile, *WOX5* can inhibit the cell division process of the quiescent center by affecting the activity of the cell cycle regulator (CYCD3) [66]. The TFs *PLT1-3 (PLETHORA 1-3)* [67,68], *SCR (SCARECROW)* [69], *BBM/PLT4 (BABYBOOM)* [70] and *SHR (SHORT ROOT)* [71,72] in the *AIL (AINTEGUMENTA-Like)* family are involved in the regulation of root apical meristem development. In Arabidopsis, plant hormones, such as cytokinin, auxin, ethylene, jasmonic acid, and gibberellin are also involved in regulating the development of root apical meristems. The information crossover between different hormone signaling pathways is important for plants to quickly respond to internal and external factors. It is also critical in performing real-time dynamic control. In addition, the expressions of some TFs that regulate the development of root apical meristems are also regulated by plant hormones [63,73,74].

### 4.2. Shoot Apical Meristem (SAM)

The plant shoot apical meristem is a hemispherical dome-like structure in which cell populations can undergo cell division to maintain self-renewal or develop into organs [75]. In the aerial parts of plants, the shoot apical meristem is responsible for producing and developing the leaves, floral meristems, and stems throughout the plant life cycle. At the histological level, the shoot apical meristem can be divided into the central, peripheral, and rib zone [76,77]. At the top of the stem-end meristem, the dome is the so-called central zone, where stem cells are located.

In the core of a complex regulatory network that determines the size and location of the central region, there is a signaling pathway involving the TF *WUS* (*WUSCHEL*), the receptor kinase CLV1 (CLAVATA 1), the receptor-like protein CLV2, and the ligand CLV3. The *WUS* controls the differentiation and maintenance of stem cells, which is expressed centrally in the tissue below the central zone. Ectopic expression of *WUS* can promote the transition of plants from vegetative to embryogenic growth [78]. Deletion of *WUS* will lead to the differentiation of stem cells and the loss of stem meristems [79]. In confirmation of previous conclusions regarding the importance of *WUS* in regulating the shoot meristem, recent reports further show that the miR156-SPL (squamosa promoter binding protein-like) pathway is directly or indirectly involved in the WUS mechanism regulating SAM size [80]. The small peptide of CLV3 expressed by stem cells negatively regulates *WUS* expression through the CLV1 receptor [81]. In the SAM, the CLV3-WUS feedback signaling affects stem cell proliferation and differentiation through an autoregulatory negative feedback loop comprising the stem cell-promoting TF WUS and the differentiation promoting peptide CLV3 [82,83]. Phytohormones are required for maintaining the homeostasis of shoot stem cells, and cross-talk exists between WUS function and cytokinin action in the SAM. Cytokinin signaling has been observed to play an important role in maintaining both shoot meristem activity and proliferation, by acting through AHK (Arabidopsis histidine kinase) receptors, which then pass on the signal to the two TF classes: ARRs (type-A Arabidopsis response regulators) and type-B ARRs [84,85,86].

### 4.3. Lateral Meristem

The meristem, which is distributed in a barrel shape on the near surface of some plant rhizome and other organs in the direction parallel to the long axis is called the lateral meristem. Lateral meristems include vascular cambium and cork cambium. Among them, plant vascular tissue is composed of xylem, phloem, and cambium/procambium, and exists radially in organs. The formation of vascular tissue patterns in Arabidopsis begins during embryonic development, along the root-hypocotyl axis, and differentiates from the precursor-procambium of vascular tissue.

The well-organized pattern of the plant vascular system is also regulated by different intercellular signals, and the regulatory mechanism of vascular system development is similar to that of shoot apical meristem and root apical meristem, both of which are controlled by plant hormones and genes. TDIF peptides (tracheary element differentiation inhibitory factor) were isolated from *Zinnia elegans*, and its C-terminal 12 amino acid motifs were identical to the conserved 12 motifs of *CLE41/44 (CLV3/ESR-related 41/44)* in Arabidopsis. The differentiation of stem cell–like procambial cells into xylem cells is inhibited by TDIF by participating in regulating the formation of vascular tissue and promoting the cell division [87]. The *CLE41* and *CLE44* genes of Arabidopsis encode TDIF, and according to previous studies on TDIF, the fate of vascular stem cells may also be controlled by the CLE receptor system [88]. TDIF/CLE41/CLE44 is expressed in the procambium/cambium and binds to the CLV1-related LRR-RLK (leucine-rich repeat receptor-like protein kinase) protein TDR/PXY (TDIF RECEPTOR/PHLOEM INTERCALATED WITH XYLEM). *WOX4* and *WOX14* are downstream factors of the TDIF/CLE41/CLE44-TDR/PXY signaling pathway and positively regulate the division of pro-cambial cells [58]. In addition, when TDR/PXY and *WOX4* genes were deleted, the expression of some ERFs was induced, indicating that during the formation of vascular tissue, there is a link between the TDIF/CLE41/CLE44-TDR/PXY and the ethylene signaling pathway [89]. Studies have shown that the TDIF/CLE41/CLE44-TDR/PXY-WOX4 signaling pathway is a conserved mechanism in regulating plant vascular tissue development and can cooperate with the ER (endoplasmic reticulum) signaling pathway [90,91,92]. The xylogen polypeptide hormone is involved in the formation of plant vascular tissue. Arabidopsis *atxyp1 atxyp2* mutants have defects in vascular tissue structure, which are manifested as discontinuous and thickened leaf veins, abnormal vascular connections, and simplified leaf vein structure [93], indicating that At XYP1 and At XYP2 proteins may play an important regulatory role in the development of vascular tissue. Rice (*Oryza sativa* L.) *xylp7* mutants have significantly shortened internode spacing except for the lowest internode [94].

## 5. Autophagy in Plant Meristems

### 5.1. Autophagy in Root Apical Meristem (RAM) under Stress Conditions

Root growth depends on apical meristem cell division, followed by cell elongation and differentiation. When plants suffer from nutrient starvation, the number of autophagosomes increases in the elongation and differentiation regions, and numerous autophagosomes appear in plant cells near the apical meristem [95]. In addition, the canonical traits of the plant-type autophagosomes-autolysosomes pathway was observed in the root tip cells that received systemic PCD induced by TMV (tobacco mosaic virus) local infection for cell contents removal and recovery [96]. *ATG* genes are required for autophagy in root tip cells. In Arabidopsis root tip cells, both ATG2 and ATG5 proteins are necessary for autophagy, while the ATG9 protein contributes to but is not required for autophagy [42]. Autophagy is involved not only in nutrient recycling under nutrient-limited conditions, but also in root hair formation and cell growth in plant root cells. Plants respond to nutrient deficiency in the soil by increasing the number of lateral roots (LR) to increase the surface area of the root. Consistent with this, phosphate availability was found to have a significant effect on the root system architecture [97].

The root meristem is responsible for optimal root growth and architecture, and its stem cell activity is sustained by a variety of factors, such as nutrient availability (e.g., glucose), hormonal levels (e.g., auxin) and ROS species homeostasis (e.g., superoxide and H_2_O_2_) [98,99,100]. Glucose produced by photosynthesis is a hormone-like signaling molecule that regulates plant development and physiological activities. Through genetic, genomic, and systematic analyses, photosynthesis-driven glucose-TOR (target of rapamycin) signaling was found to stimulate and maintain meristem activity for unlimited root growth [101]. In plants, TOR can integrate multiple exogenous and endogenous signals to coordinate several downstream processes, such as cell division and elongation, nutrient transport and metabolism, biological rhythm, and stress response [102]. In addition to regulating root development, glucose-TOR also promotes hypocotyl elongation. Studies have shown that under dark conditions, glucose-TOR signaling stabilizes the key TF BZR1 in the BRs signaling pathway by inhibiting the autophagy process, thereby promoting hypocotyl elongation [103]. After further studies found that autophagy-deficient mutants exhibit greater tolerance to glucose, and accumulate ROS, induced by less glucose at the root tips, Arabidopsis sensors may signal the constitutive autophagy system through the direct or indirect action of TOR after sensing high glucose stress in roots; these results indicate that autophagy regulates the activity of glucose-mediated root meristems in Arabidopsis by modulating ROS production [104] (Figure 2). TOR signaling can modulate glucose-suppressed root meristem activity and negatively regulates autophagy in plants [98,105]. Low levels of ROS, which are produced as byproducts of cellular metabolism, act as critical secondary messengers, controlling a number of vital developmental processes, such as root meristem maintenance [101,106]. However, excessive ROS may cause severe oxidative damage to cells. ROS are actively produced in the root tips, and their concentrations control the extent and direction of root growth [107].

In Arabidopsis mutants *abo6* (*ABA overly sensitive 6*) and *abo8* (*ABA overly sensitive 8*), AT5G04895/ABO6 and AT4G11690/ABO8 lack a pentatricopeptide repeat protein and DExH box RNA helicase, respectively, resulting in increased ABA sensitivity and decreased meristem activity in roots [108,109]. In addition, the root meristem phenotype in these mutants could be partially rescued by the addition of the reducing agent glutathione (GSH), suggesting that mitochondrial ROS in the root tip is a crucial retrograde signal to maintain its meristem activity [108,109]. A large number of studies have shown that ROS from different sources can regulate autophagy in plant cells. Exogenous H_2_O_2_ treatment causes severe oxidative stress in Arabidopsis, which induces the process of autophagy [110]. In Arabidopsis *atg2* and *atg5* mutants, massive accumulation of H_2_O_2_ was observed [21], similar to the results of H_2_O_2_ treatment. ROS plays a key role in maintaining root meristem activity, and deletion mutations in ATGs lead to ROS accumulation. Thus, it can be seen that autophagy and ROS may regulate root meristem activity, but how the two co-regulate this process is still unknown. Besides glucose and ROS, auxin is an important hormone that regulates root growth and plays a crucial role in maintaining the root meristem [111,112]. Auxin accumulates in primary root tips via the action of polar transporters, such as the pin-formed proteins (PINs), and it contributes to the root patterning and helps in regulating the root cell division [113]. Recent evidence indicates a link between regulating auxin-dependent lateral root development and autophagy under the conditions of phosphate starvation in Arabidopsis. Under phosphate-deficient conditions, LR development and auxin accumulation in root meristems are inhibited when autophagy is inhibited by treatment with the autophagy inhibitor 3-methyladenine, indicating that autophagy plays an important role in regulating LR development [114,115] (Figure 2).

### 5.2. Autophagy in Shoot Apical Meristem (SAM) under Stress Conditions

The cells at the tip of the stem, in the SAM, are responsible for deriving the aerial structure of higher plants. The SAM produces lateral organs and stem tissues and also regenerates itself throughout the plant life. Therefore, development of plant under stress is dependent on the meristem, and the SAM of seedlings plays a crucial role in plant defense signaling. Ultrastructural changes in the shoot apical meristem and formation of autophagic vacuoles in canola (*Brassica napus cv*. Symbol) have been observed under salinity conditions [116]. A recent study revealed that autophagy was involved in thermopriming-induced defense mechanism in SAM of young Arabidopsis. Most of the *ATG* expressions were increased, and autophagy was induced after the thermopriming treatment which caused autophagosome formation in the shoot apices [117]. This means autophagy is recommended as a potential protection system for the homeostasis of SAM.

The relationship between plant autophagy and deficiency of micronutrients in SAM has recently been discovered in Arabidopsis. Although nutrient deficiency interferes with plant growth, an excess of certain elements can also be problematic. The concentration of some important nutrients in the soil environment must remain within a certain range for healthy growth, and the optimal range for micronutrients is narrower than for macronutrients. Plant growth can be inhibited by excess Zn in the rhizosphere. Plants display growth defects in both roots and shoots, and chlorosis in shoots occurs when Zn is in excess. Excessive Zn competes with other divalent metal elements, which is one of its toxic effects. Due to the fact that divalent metal ions are transported into plant cells via the same transporter, excess Zn can result in a deficiency of other divalent metal ions [118]. In the presence of Zn excess, autophagy is activated, and autophagy-deficient Arabidopsis plants exhibit obvious growth defects and chlorosis. However, the excess supply of Fe markedly alleviates growth defects and chlorosis due to an excess of Zn, implying that symptoms caused by excess Zn in *atg* mutants are due to Fe starvation. In consequence, autophagy has been discussed in the distribution of Fe^3+^ to juvenile leaves to promote healthy plant growth in the presence of excess Zn [119].

### 5.3. Autophagy in Other Types of Meristems under Stress Conditions

The function of autophagy in other plant meristems has also been revealed. The study performed in potato (*Solanum tuberosum*) tuber apical bud meristem (TAB-meristem) revealed the role of autophagy in controlling cold-induced apical dominance [120] (Table 1). GAPC1, GAPC2, and GAPC3 have the ability to interact with ATG3 both in vitro and in vivo in potatoes. In plants, the interaction between GAPC and *ATG3* negatively regulates autophagy [121]. Therefore, cell death in TAB-meristem could be regulated by the ATG3-dependent process. The interactions between *ATG3* and GAPCs could be an important factor for the apical dominance of potato tubers [120]. By combining the data from RNA-seq analysis and ERF-VII targeted analysis, it was found that bud populations on leafless vines are gradually starved during natural dormancy cycles, and that starvation-induced ABA acts to temporarily inhibit meristem growth [118]. While starvation-triggered autophagy may be the primary catabolic resource, renewal of energy availability induces degradation of ABA and restoration of meristem activity [33]. In wheat, basal florets are gradually approaching the fertile stage while distal floret rot is observed. In studying the mechanism of fertile flower number in wheat, autophagy was found to play a role in whole flower decay as a mechanism for regulating the number of fertile florets [122]. However, there is no direct evidence that this phenomenon is caused by the effect of autophagy on the floral meristem.

## 6. Concluding Remarks and Future Perspectives

The differentiation of plant meristems is an extremely complex process, involving a large number of morphological and structural changes, environmental factors, endogenous hormone and molecular regulatory mechanisms. Previous studies on the regulation mechanism of meristem differentiation mainly focused on hormone and transcriptional regulation. In recent years, with the significant increase in plant autophagy research, many functions of autophagy in regulating meristems have received more attention. To date, most of the studies focus on the relationship between autophagy and apical meristem, and it has preliminarily clarified two important aspects of autophagy on apical meristem. First, regulation of meristem activity under different energy supply conditions. Second, it protects meristems from environmental stresses. However, little is known about whether autophagy can directly regulate other meristems. The relationship between autophagy and other meristems remains to be further explored. Although the regulation of meristems by autophagy has been established, the underlying molecular mechanisms remain unclear. Therefore, the functional research on autophagy in the process of meristem differentiation should be strengthened to elucidate the regulatory mechanism of autophagy more specifically. In addition, current research mainly focuses on the connection between autophagy and meristems, but many studies have found that phytohormones regulate meristems and autophagy. The relationship between phytohormones, meristems, and autophagy remains to be further explored. Exploring these mechanisms will help us gain more comprehensive understanding of autophagy physiological processes.

## Figures and Tables

**Figure 1 ijms-23-06236-f001:**
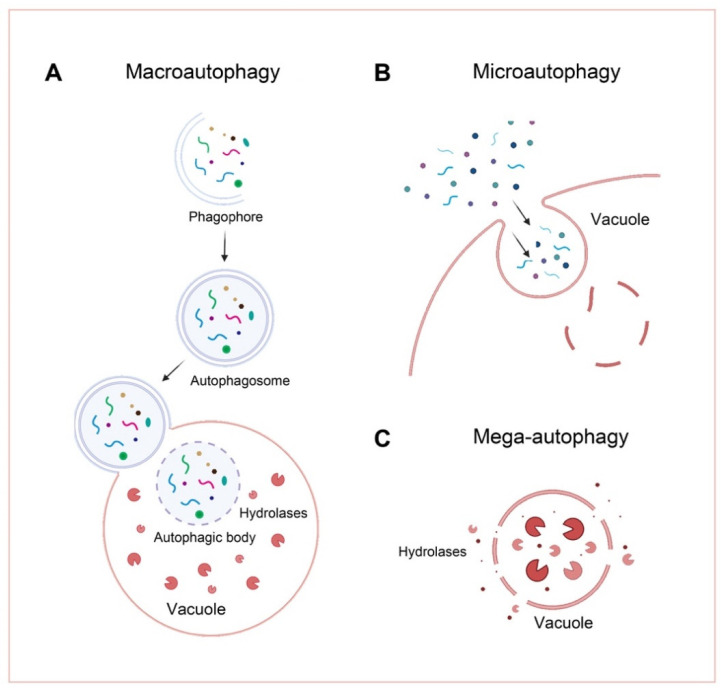
Schematic of the three types of plant autophagy. (**A**) In macroautophagy, a cup-shaped phagophore is formed and closes into double-membrane autophagosomes while wrapping the cellular substances. The autophagosome is then delivered to the vacuole, fused with the tonoplast, and degraded in the vacuolar lumen. (**B**) In microautophagy, intracellular material directly enters the vacuole by vacuolar endocytosis and is degraded. (**C**) During mega-autophagy, the tonoplast permeabilizes and ruptures, releasing a large number of hydrolases into the cytoplasm, resulting in indiscriminate degradation of cytoplasmic materials.

**Figure 2 ijms-23-06236-f002:**
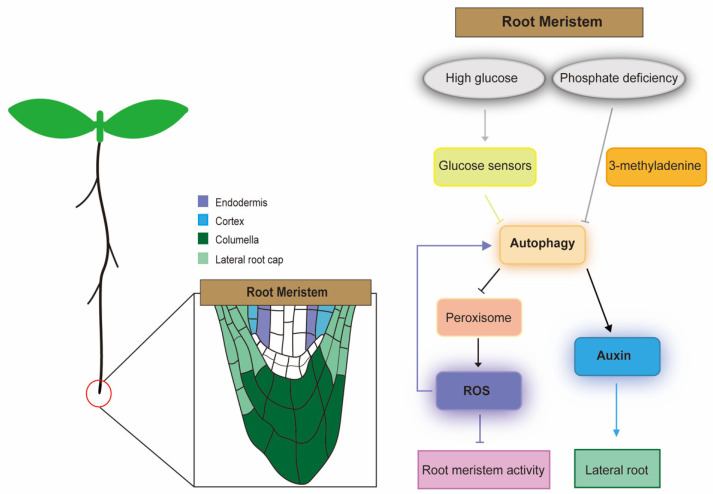
Schematic diagram of the regulatory network of autophagy in the Arabidopsis root meristem. Under high glucose stress, the glucose sensor signals to the autophagy system. Autophagy further regulates peroxisomes and contributes to the production of ROS. ROS accumulation under the induction of high levels of glucose impairs root meristem activity, and high levels of ROS also enhance autophagy mechanisms to maintain root meristem function under stress conditions. Under the phosphate-deficient condition, LR development and auxin accumulation in root meristems are inhibited when autophagy is inhibited by treatment with the autophagy inhibitor 3-methyladenine. Arrow indicates positive interactions; barred arrows indicate repressive interactions.

**Table 1 ijms-23-06236-t001:** Autophagy-linked proteins in plant meristems.

Meristem Type	Protein/s	Species	Experimental Condition	Function of Protein/s	Reference
Root meristem	AtATG2, AtATG5, and AtATG9	Arabidopsis (*Arabidopsis thaliana*)	Normal growth conditions and sucrose starvation	Autophagy	[42]
Root meristem	PsCBL and PsCIPK	Arabidopsis	Nutrient-sufficient and starvation conditions	Calcium and stress signals	[95]
Root meristem	ARK2 and AtPUB9	Arabidopsis	Phosphate starved conditions	Lateral root (LR) development	[114]
Root meristem	ARK2 and AtPUB9	Arabidopsis	Phosphate starved conditions	LR development	[115]
Root meristem	AtATG5 and AtATG7	Arabidopsis	Various concentrations of glucose conditions	Peroxisome	[104]
Shoot meristem	AtATG8	Arabidopsis	Thermopriming treatment	Capital ATGs in Shoot apical meristem (SAM)	[117]
Shoot meristem	AtATG2 and AtATG5	Arabidopsis	Excess Zn conditions	Autophagy	[119]
Tuber apical bud meristem	StGAPC1, StGAPC2, and StGAPC3	Potato (*Solanum tuberosum* L.)	Normal conditions	Diverse physiological and developmental processes	[120]
Bud meristem	VvERF057 and VvERF059	Grape (*Vitis vinifera cv.* Early sweet)	Chemical and physical treatments	Energy-regenerating	[123]

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
