# Peer review of "Autophagy-Mediated Regulation of Different Meristems in Plants"

_ijms, 2022, doi:10.3390/ijms23116236_

Round 1

Reviewer 1 Report

Autophagy in plants is a relatively recent reseach area, and the molecular bases of this process are not fully understood. Therefore, I do consider interesting to deal with this topic and to contribute to spreading the existing knowledge among readers of IJMS.  

However, I would like to make some suggestions to the authors, wich I hope migh help to improve the manuscript in the current form.

  1. Avoid repeating words in upcoming sentences and even within the same sentence (i.e. line 16-17…endogenous hormonal and molecular regulation…
  2. Avoid saying general sentences without letting clear the idea or the meaning enclosed (i.e. in the same sentence, “and shows obvious changes”. What do you exactly want to say? This consideration should be taken into account throughout the ms.
  3. Given that the title focusses on the autophagy-mediated regulation of different meristems, I would suggest to pay attention to the length of the manuscritp coping with it, and which represents approximately one fifth of the total. I would suggest to provide more detail in those paragraphs connecting with the main purpose of the review.
  4. I would suggest to write the complete name of genes and proteins, followed by the abreviation in parentheses, and also to give the meaning of each abreviation the first time they are mentioned in the text. Line 35 ATG8…Autophagy gene 8. Besides, genus name is mostly writen in italic as far as I know.
  5. Regarding the content of those parts related to the role of autophagy in different plant developmental processes, I would suggest to present them in more detail, as the bibliography have been revised quite extensively. I found an excess of information in some aspects actually better known, and a lack of information about the role of autophagy either in growth and development or when plants are under the effect of abiotic and biotic stress.
  6. Line 30. Hypoxial and
  7. Line 37. …steps of plant reproduction such as controlling….
  8. Line 41. Write the name of the phytohormone before the abbreviation, i.e. salicilic acid (SA), when mentioned first time.
  9. Line 42. When it is said…”The accumulation of SA is responsible for the early leaf senescence”, I consider you should be more precise. It seems all the weight is carried out by SA but I have serious doubts.
  10. Line 168. Perhaps organelles should be replaced by other words as structure.
  11. Line 179. I have never heard endothelium but endoderms.
  12. Line 195. Please add some reference to the last sentence.
  13. Line 201. Please, avoid repeating zone, better…Is divided into the central, peripheral and rib cell zones.
  14. Line 213. Please, replace by…repressing the expression of WUS, in order…
  15. Line 247. Please, replace by…Between the TDIF…..and ethylene signaling pthway.
  16. Line 344. Please, revise the legend of the table. You should mention that they are proteins linked to autophagy. What about “Autophagy-linked proteins in plant meristems”?
  17. Line 345. Concluding remarks are not well arranged. I would remove from line 346 to A recent study…in the line 351. Also, it is not needed to repeat the types of meristems here. Also, ne 264. Please, remove but in the line 264.
  18. Finally, I found the figure 1 very sample to be included in a review of 2022. Maybe it would look better referring to molecular pathways, or something like that.

Author Response

Reviewer#1

Comments and Suggestions for Authors

Autophagy in plants is a relatively recent reseach area, and the molecular bases of this process are not fully understood. Therefore, I do consider interesting to deal with this topic and to contribute to spreading the existing knowledge among readers of IJMS.  

Response: Thank you so much. According to your suggestions, overall manuscript has been improved. Moreover, the position of all the sections were changed according to the suggestions of one of the reviewers.

However, I would like to make some suggestions to the authors, wich I hope migh help to improve the manuscript in the current form.

  1. Avoid repeating words in upcoming sentences and even within the same sentence (i.e. line 16-17…endogenous hormonal and molecular regulation…

Response: Thank you so much, repetitions have been deleted.

  1. Avoid saying general sentences without letting clear the idea or the meaning enclosed (i.e. in the same sentence, “and shows obvious changes”. What do you exactly want to say? This consideration should be taken into account throughout the ms.

Response: Thanks, this sentence has been changed.

  1. Given that the title focusses on the autophagy-mediated regulation of different meristems, I would suggest to pay attention to the length of the manuscritp coping with it, and which represents approximately one fifth of the total. I would suggest to provide more detail in those paragraphs connecting with the main purpose of the review.

Response: Thanks, we have added some details.

  1. I would suggest to write the complete name of genes and proteins, followed by the abreviation in parentheses, and also to give the meaning of each abreviation the first time they are mentioned in the text. Line 35 ATG8…Autophagy gene 8. Besides, genus name is mostly writen in italic as far as I know.

Response: Thank you so much, it has been revised.

  1. Regarding the content of those parts related to the role of autophagy in different plant developmental processes, I would suggest to present them in more detail, as the bibliography have been revised quite extensively. I found an excess of information in some aspects actually better known, and a lack of information about the role of autophagy either in growth and development or when plants are under the effect of abiotic and biotic stress.

Response: Thank you so much, more details have been added.

  1. Line 30. Hypoxial and

Response: Thank you so much, it has been revised.

  1. Line 37. …steps of plant reproduction such as controlling….

Response: Thank you so much, it has been revised.

  1. Line 41. Write the name of the phytohormone before the abbreviation, i.e. salicilic acid (SA), when mentioned first time.

Response: Thank you so much. According to your suggestion, changes have been made.

  1. Line 42. When it is said…”The accumulation of SA is responsible for the early leaf senescence”, I consider you should be more precise. It seems all the weight is carried out by SA but I have serious doubts.

Response: Thanks, this sentence has been modified.

  1. Line 168. Perhaps organelles should be replaced by other words as structure.

Response: Thank you so much, it has been revised.

  1. Line 179. I have never heard endothelium but endoderms.

Response: Thank you so much, it has been corrected.

  1. Line 195. Please add some reference to the last sentence.

Response: Thank you so much, we have added some references.

  1. Line 201. Please, avoid repeating zone, better…Is divided into the central, peripheral and rib cell zones.
  1. Response: Thanks, repetitions have been deleted.
  1. Line 213. Please, replace by…repressing the expression of WUS, in order…

Response: Thank you so much, the sentence has been rewritten; please check indicated lines.

  1. Line 247. Please, replace by…Between the TDIF…..and ethylene signaling pthway.

Response: Thank you so much, it has been replaced.

  1. Line 344. Please, revise the legend of the table. You should mention that they are proteins linked to autophagy. What about “Autophagy-linked proteins in plant meristems”?

Response: Thanks, we have modified it according to your suggestion.

  1. Line 345. Concluding remarks are not well arranged. I would remove from line 346 to A recent study…in the line 351. Also, it is not needed to repeat the types of meristems here. Also, ne 264. Please, remove but in the line 264.

Response: Thank you so much, the line 346 to line 351 has been deleted. Changes have been made according to your suggestion.

  1. Finally, I found the figure 1 very sample to be included in a review of 2022. Maybe it would look better referring to molecular pathways, or something like that.

Response: Thank you so much. This article mainly focuses on the relationship between meristems and autophagy, not autophagy itself. Therefore, figure 1 only briefly describes the types of plant autophagy. For that reason, we did not add molecular pathways.

TRANSLATE with x English
Arabic Hebrew Polish
Bulgarian Hindi Portuguese
Catalan Hmong Daw Romanian
Chinese Simplified Hungarian Russian
Chinese Traditional Indonesian Slovak
Czech Italian Slovenian
Danish Japanese Spanish
Dutch Klingon Swedish
English Korean Thai
Estonian Latvian Turkish
Finnish Lithuanian Ukrainian
French Malay Urdu
German Maltese Vietnamese
Greek Norwegian Welsh
Haitian Creole Persian  
TRANSLATE with COPY THE URL BELOW Back EMBED THE SNIPPET BELOW IN YOUR SITE Enable collaborative features and customize widget: Bing Webmaster Portal Back

Reviewer 2 Report

The review entitled "Autophagy-mediated Regulation of Different Meristems in Plants" by Cheng et al. is focusing on the autophagy and different plant meristems and highlight how autophagy regulates meristems under stress conditions in plants.

This is a concise and well researched review article. Overall, I commend the authors for compiling all this information. The article provides a general overview of autophagy and plant meristems and discusses in more details how autophagy can regulate meristem homeostasis in different environmental conditions.

However, some other aspects as activation of autophagy by the energy sensor SnRK1 and inhibition by TOR kinase are not covered. Even though there are not many reports on the relationship between autophagy activation by SnRK1 and meristem regulation by SnRK1 antagonist, target of rapamycin (TOR), it would be much better if some possible connections among them are mentioned and discussed. I suppose a few comments on the recent progress would be helpful for readers

Some gentle editing for English usage is needed

line 30. Correct to "hypoxia [11]"

line 37. Please edit this sentence "[14], such as"

line 42. Should this be opposite?

"up-regulation of AS biosynthesis and accumulation in autophagy-defective mutants" You may add particularly under low nitrate conditions

line 62. Change to "mega-autophagy"

lines 76 and 77. The full name of ‘DRE- and ETH-‘ seems off, please confirm what DRE and ETH stands for

line 83. Please edit this sentence

line 88. Edit this sentence

line 132. Correct the sentence

line 182. Change to WOX5 (WUSCHEL…..)

line 213-214. Please delete ‘expression of’

edit this sentence starting at "which conform…"

line 217-218. Please edit this sentence starting at "in the…"

line 235. You may add that CLV3 and CLE41/44 belong to a large family CLE genes in Arabidopsis as well

lines 235-237. Edit this sentence starting at "mesophyll cells into vascular molecules ?…."

Author Response

Reviewer#2

Comments and Suggestions for Authors

The review entitled "Autophagy-mediated Regulation of Different Meristems in Plants" by Cheng et al. is focusing on the autophagy and different plant meristems and highlight how autophagy regulates meristems under stress conditions in plants.

This is a concise and well researched review article. Overall, I commend the authors for compiling all this information. The article provides a general overview of autophagy and plant meristems and discusses in more details how autophagy can regulate meristem homeostasis in different environmental conditions.

However, some other aspects as activation of autophagy by the energy sensor SnRK1 and inhibition by TOR kinase are not covered. Even though there are not many reports on the relationship between autophagy activation by SnRK1 and meristem regulation by SnRK1 antagonist, target of rapamycin (TOR), it would be much better if some possible connections among them are mentioned and discussed. I suppose a few comments on the recent progress would be helpful for readers

Some gentle editing for English usage is needed

 Response: Thank you very much for your helpful suggestions and valuable input in our manuscript. According to your suggestions, overall manuscript has been revised carefully. A point-by-point response is provided below. The revisions are highlighted in the main text with track changes.

line 30. Correct to "hypoxia [11]"

Response: Thank you so much, it has been corrected.

 line 37. Please edit this sentence "[14], such as"

Response: Thank you so much, this sentence has been edited.

line 42. Should this be opposite?

"up-regulation of AS biosynthesis and accumulation in autophagy-defective mutants" You may add particularly under low nitrate conditions

Response: Thank you so much, it has been added.

line 62. Change to "mega-autophagy"

Response: Thank you so much, it has been changed.

lines 76 and 77. The full name of ‘DRE- and ETH-‘ seems off, please confirm what DRE and ETH stands for

 Response: Thank you so much, it has been revised.

line 83. Please edit this sentence

Response: Thank you so much, this sentence has been edited.

line 88. Edit this sentence

Response: Thank you so much, this sentence has been edited.

line 132. Correct the sentence

 Response: Thank you so much, this sentence has been corrected.

line 182. Change to WOX5 (WUSCHEL…..)

 Response: Thank you so much, it has been changed.

line 213-214. Please delete ‘expression of’

edit this sentence starting at "which conform…"

 Response: Thank you so much, this sentence has been edited.

line 217-218. Please edit this sentence starting at "in the…"

 Response: Thank you so much, this sentence has been edited.

line 235. You may add that CLV3 and CLE41/44 belong to a large family CLE genes in Arabidopsis as well

 Response: Thank you so much, it has been added.

lines 235-237. Edit this sentence starting at "mesophyll cells into vascular molecules ?…."

 Response: Thank you so much, this sentence has been edited.

TRANSLATE with x English
Arabic Hebrew Polish
Bulgarian Hindi Portuguese
Catalan Hmong Daw Romanian
Chinese Simplified Hungarian Russian
Chinese Traditional Indonesian Slovak
Czech Italian Slovenian
Danish Japanese Spanish
Dutch Klingon Swedish
English Korean Thai
Estonian Latvian Turkish
Finnish Lithuanian Ukrainian
French Malay Urdu
German Maltese Vietnamese
Greek Norwegian Welsh
Haitian Creole Persian  
TRANSLATE with COPY THE URL BELOW Back EMBED THE SNIPPET BELOW IN YOUR SITE Enable collaborative features and customize widget: Bing Webmaster Portal Back

Round 2

Reviewer 1 Report

After reading the new version of the manuscript, I welcome many suggestions were incorporated. Saying that, there are some additional corrections the authors should still modify, and which are summarised as follows:

1.       Line 32. Please, choose one option to finish the sentence…hypoxia, high salt, etc. or hypoxia and high salt.

2.       Line 32. Please, revise the sentence…A basal level of autophagy maintained when plant are growing/ grow under normal condition is…

3.       Line 43. Metabolic or metabolomic? and what are you talking about? Please, explain a bit more.

4.       Line 56. Meristems vs development of plant meristems sounds not so clear to me.

5.       Lines 57-59. Role…role…please, avoid repeat words placed near.

6.       Line 62. I think is not necessary to repeat the definition of Autophagy. Better to put paragraph 2 to detail the types of Autophagy. Then, 2.1. Macroautophagy; 2.2. microautophagy

7.       Regarding mega-autophagy, I have some doubts about the difference with plant apoptosis. Could you please clarify this point?

8.       Paragraph starting in the line 126 should be better structured and not jumping from one issue to another, without any connection.

9.       Line 145. Please, remove “during seed germination”.

10.   Line starting in 145. Please, be more explicit with the idea presents in the reference 45.

11.   Line 156. In the process of plant growth and development…What were you talking about before? They were all developmental processes too.

12.   Line 156. Is there no references after the 146 about the formation of xyleme in Populus? Please, extend the idea quite a bit.

13.   Line 166. Put ”:” before the term second ROS

14.   Line 177. What is chlorophagy?

15.   Line 177. Oversupply….supply…please avoid repeat words

16.   Line 182. Please, put SA when mention salicylic acid as it was already detailed.

17.   Line 187. MdATG…please mention the species.

18.   Line 195. Please, put B. cinerea second time. Then, it is not so clear the idea regarding the relationship between autophagy and different strategies in defence.

19.   The division of meristems is not correct. There are embryonary meristems (SAM and RAM); postembryonary meristems, linked to primary growth (axilar, intercalar, marginal) and secondary growth (cambium and phelloderm), and there are also flowering meristems. In fact, you mention floral meristems some times.

20.   Line 212. Please, use the word “endodermis”.

21.   Line 236. Please, replace “which is” by “there is”.

22.   Line 242. Please, replace by “from vegetative to embryonic growth”

23.   Line 249. Please, join SQUAMOSA

24.   Line 264. Lateral meristem. What do you refer to? The content is about the formation of vascular tissue.

25.   Please, be consistent with the use of Arabidopsis or A. thaliana throughout the ms.

26.   Line 311. It is not necessary to put At before ATG2 and ATG5, as you have already noticed A. thaliana.

27.   Line 362. Please, explain the idea turning around thermoprimming, and sucessive text, including what happens with the divalent ions as you mention Fe+, which is trivalent.

28.   Line 385. After mentioning potato (Solanum tuberosum) there is no need to put At before the genes.

29.   I do recommend to improve the concluded remarks, giving them more consistency with the previous statements provided along the text. There is no need to repeat basic concepts but briefly highlighting the most important achievements and  expected goals in the future.

30.   Finally, I would suggest to connect better each idea showed in each paragraph, avoiding to give the impresion they are isolated thoughts, forgetting the plant development framework in which they are involved.

Author Response

Reviewer#1

Comments and Suggestions for Authors

After reading the new version of the manuscript, I welcome many suggestions were incorporated. Saying that, there are some additional corrections the authors should still modify, and which are summarised as follows:

  1. Line 32. Please, choose one option to finish the sentence…hypoxia, high salt, etc. or hypoxia and high salt.

Response: Thank you so much, it has been revised.

  1. Line 32. Please, revise the sentence…A basal level of autophagy maintained when plant are growing/ grow under normal condition is…

Response: Thank you so much. According to your suggestions, it has been revised.

  1. Line 43. Metabolic or metabolomic? and what are you talking about? Please, explain a bit more.

Response: Thank you so much, the correct spelling is metabolomics. It has been explained in the sentence.

  1. Line 56. Meristems vs development of plant meristems sounds not so clear to me.

Response: Thank you so much, it has been revised.

  1. Lines 57-59. Role…role…please, avoid repeat words placed near.

Response: Thank you so much, it has been revised.

  1. Line 62. I think is not necessary to repeat the definition of Autophagy. Better to put paragraph 2 to detail the types of Autophagy. Then, 2.1. Macroautophagy; 2.2. microautophagy

Response: Thank you so much. According to your suggestion, it has been revised.

  1. Regarding mega-autophagy, I have some doubts about the difference with plant apoptosis. Could you please clarify this point?

Response: Thank you so much, mega-autophagy by permeabilization or rupture of the lysosome or tonoplast.This permeabilization appears to be common in plant PCD. It results in the release of vacuolar hydrolases, which can degrade whatever is left in the cell. Tonoplast permeabilization and the subsequent rapid disappearance of the cellular contents has been reported in detailed investigations of tracheary element (xylem) PCD (Obara, 2001). It has been observed in PCD during aerenchyma formation (Evans, 2001; Drew, 2000), phloem cell development, formation of root cap cells, and during the senescence of chloroplast-containing cells, such as in leaf cells (Gahan, 1982).

  1. Paragraph starting in the line 126 should be better structured and not jumping from one issue to another, without any connection. 

Response: Thank you so much, it has been revised.

  1. Line 145. Please, remove “during seed germination”.

Response: Thank you so much, it has been removed.

  1. Line starting in 145. Please, be more explicit with the idea presents in the reference 45.

Response: Thanks, it has been added.

  1. Line 156. In the process of plant growth and development…What were you talking about before? They were all developmental processes too.

Response: Thanks, it has been changed.

  1. Line 156. Is there no references after the 146 about the formation of xyleme in Populus? Please, extend the idea quite a bit.

Response: Thank you so much, it has been added.

  1. Line 166. Put ”:” before the term second ROS

Response: Thank you so much, it has been changed.

  1. Line 177. What is chlorophagy?

Response: Thank you so much.

The degradation of chloroplasts via autophagy, namely chlorophagy, consists of two pathways: the whole chloroplast pathway and the RCB pathway (Otegui, 2018; Izumi, 2019; Zhuang and Jiang, 2019).

  1. Line 177. Oversupply….supply…please avoid repeat words

Response: Thank you so much, it has been revised.

  1. Line 182. Please, put SA when mention salicylic acid as it was already detailed.

Response: Thank you so much, it has been revised.

  1. Line 187. MdATG…please mention the species.

Response: Thank you so much, it has been mentioned inThe overexpression of MdATG18a increased the basal heat tolerance of apple (Malus domestica) plants”.

  1. Line 195. Please, put B. cinereasecond time. Then, it is not so clear the idea regarding the relationship between autophagy and different strategies in defence.

Response: Thank you so much, we have modified it according to your suggestion.

  1. The division of meristems is not correct. There are embryonary meristems (SAM and RAM); postembryonary meristems, linked to primary growth (axilar, intercalar, marginal) and secondary growth (cambium and phelloderm), and there are also flowering meristems. In fact, you mention floral meristems some times.

Response: Thank you so much. According to your suggestion, it has been modified.

  1. Line 212. Please, use the word “endodermis”.

Response: Thank you so much, it has been revised.

  1. Line 236. Please, replace “which is” by “there is”.

Response: Thank you so much, it has been revised.

  1. Line 242. Please, replace by “from vegetative to embryonic growth”

Response: Thank you so much, it has been revised.

  1. Line 249. Please, join SQUAMOSA

Response: Thank you so much, it has been revised.

  1. Line 264. Lateral meristem. What do you refer to? The content is about the formation of vascular tissue.

Response: Thank you so much, lateral meristems include vascular cambium and cork cambium. We have added a description to the text.

  1. Please, be consistent with the use of Arabidopsis or A. thaliana throughout the ms.

Response: Thank you so much, it has been revised.

  1. Line 311. It is not necessary to put At before ATG2 and ATG5, as you have already noticed A. thaliana.

      Response: Thank you so much, it has been deleted.

  1. Line 362. Please, explain the idea turning around thermoprimming, and sucessive text, including what happens with the divalent ions as you mention Fe+, which is trivalent.

Response: Thank you so much, thermopriming which is a promoted acquired resistance stimulus. This work showed that by a mild stress (thermopriming), autophagy as one of its subsequent signaling networks has been activated in plant apical meristem and relevant thermopriming-induced autophagy genes were identified (Taheri, 2021).

Autophagy degrades intracellular structures to release labile Fe3+. Iron is converted from Fe3+ to Fe2+ by FRO2 (ferric-chelate reductase), then Fe2+ is taken up by IRT1, an iron and zinc transporter (Fukao, 2011; Daiki, 2021).

  1. Line 385. After mentioning potato (Solanum tuberosum) there is no need to put At before the genes.

Response: Thank you so much, it has been revised.

  1. I do recommend to improve the concluded remarks, giving them more consistency with the previous statements provided along the text. There is no need to repeat basic concepts but briefly highlighting the most important achievements and  expected goals in the future.

 Response: Thank you so much, according to your suggestions, changes have been made.

  1. Finally, I would suggest to connect better each idea showed in each paragraph, avoiding to give the impresion they are isolated thoughts, forgetting the plant development framework in which they are involved.

Response: Thanks, it has been revised.

TRANSLATE with x English
Arabic Hebrew Polish
Bulgarian Hindi Portuguese
Catalan Hmong Daw Romanian
Chinese Simplified Hungarian Russian
Chinese Traditional Indonesian Slovak
Czech Italian Slovenian
Danish Japanese Spanish
Dutch Klingon Swedish
English Korean Thai
Estonian Latvian Turkish
Finnish Lithuanian Ukrainian
French Malay Urdu
German Maltese Vietnamese
Greek Norwegian Welsh
Haitian Creole Persian  
TRANSLATE with COPY THE URL BELOW Back EMBED THE SNIPPET BELOW IN YOUR SITE Enable collaborative features and customize widget: Bing Webmaster Portal Back

Round 3

Reviewer 1 Report

Plase, find some additinal advises, hoping you find them useful.

Line 59. I can not understand the meaning of this statement. In the table 1, what do you mean with “other meristems”? Please, use TAB-meristem for tuber apical bud meristem , etc.

-Please, change the numbers in the paragraphs by 2, 2.1; 2.2.

-Line 319. Why do you mention “rhizome” ? It is an underground shoot. I do not understand the usefulness in the current plant context, sorry.

-Line 394. …I can not understand the entire paragrah. They are not clear the ideas. All is confused: glucose high, low, stress, TOR signaling, ROS….please, write meaningful and connected senteces.

-Line 441 and following….Autophagy in shoot apical meristem (SAM). I have not clear what are you talking about. You stat talking about stress. Perhaps you should rename the paragrah “Autophagy in shoot apical meristem (SAM) under stress conditions”???. Then, you talk about flowering somewho but not flower meristem…and I can not percieve the relationship among the writing and autophagy in meristems. I insist that there are long paragraphs referring to the clasical concepts of meristems and weak information deepening on autophagy...in meristems.

Author Response

Reviewer#1

Comments and Suggestions for Authors

Plase, find some additinal advises, hoping you find them useful.

Line 59. I can not understand the meaning of this statement. In the table 1, what do you mean with “other meristems”? Please, use TAB-meristem for tuber apical bud meristem , etc.

Response: Thank you so much, this sentence has been revised.

The content of table 1 has also been revised.

-Please, change the numbers in the paragraphs by 2, 2.1; 2.2.

Response: Thanks, it has been changed.

-Line 319. Why do you mention “rhizome” ? It is an underground shoot. I do not understand the usefulness in the current plant context, sorry.

Response: Thank you so much, the reason for mentioning rhizomes is to give an example of the distribution of lateral meristems in some plants for better understanding.

-Line 394. …I can not understand the entire paragrah. They are not clear the ideas. All is confused: glucose high, low, stress, TOR signaling, ROS….please, write meaningful and connected senteces.

Response: Thank you so much, we have modified it according to your suggestion.

-Line 441 and following….Autophagy in shoot apical meristem (SAM). I have not clear what are you talking about. You stat talking about stress. Perhaps you should rename the paragrah “Autophagy in shoot apical meristem (SAM) under stress conditions”???. Then, you talk about flowering somewho but not flower meristem…and I can not percieve the relationship among the writing and autophagy in meristems. I insist that there are long paragraphs referring to the clasical concepts of meristems and weak information deepening on autophagy...in meristems.

Response: Thank you so much, the paragraph has been renamed.

Floral meristems are formed from the inflorescence meristem and generate floral organs. Based on previous conclusions that autophagy regulates the number of fertile florets, we speculate whether autophagy affects meristems to affect flower development.

Since there are few related studies, we have done our best to describe the relationship between meristems and autophagy. This also indicates that there is a lot of space to explore in the research direction of autophagy regulation of meristem.

TRANSLATE with x English
Arabic Hebrew Polish
Bulgarian Hindi Portuguese
Catalan Hmong Daw Romanian
Chinese Simplified Hungarian Russian
Chinese Traditional Indonesian Slovak
Czech Italian Slovenian
Danish Japanese Spanish
Dutch Klingon Swedish
English Korean Thai
Estonian Latvian Turkish
Finnish Lithuanian Ukrainian
French Malay Urdu
German Maltese Vietnamese
Greek Norwegian Welsh
Haitian Creole Persian  
TRANSLATE with COPY THE URL BELOW Back EMBED THE SNIPPET BELOW IN YOUR SITE Enable collaborative features and customize widget: Bing Webmaster Portal Back